# Modeling, Kinematic Characteristics Analysis and Experimental Testing of an Elliptical Rotor Scraper Pump

**Zhen Zhang** [1,2], **Tiezhu Zhang** [1,2], **Hongxin Zhang** [1,2,*], **Jian Yang** [1,2], **Yang Cao** [1,2], **Yong Jiang** [1,2] **and Dong Tian** [1,2]

1    College of Mechanical and Electrical Engineering, Qingdao University, Qingdao 266071, China;
     rexzz9916@163.com (Z.Z.); zhangtz@sdut.edu.cn (T.Z.); yangxiaoming8533@163.com (J.Y.);
     cy082820@163.com (Y.C.); qdjy2008@163.com (Y.J.); Tiandong0911@126.com (D.T.)
2    Power Integration and Energy Storage Systems Engineering Technology Center (Qingdao),
     Qingdao 266071, China
*    Correspondence: qduzhx@126.com; Tel.: +86-135-7386-5229

**Abstract:** Against the backdrop of the rapid burgeoning of international energy engineering, the efficient utilization of energy is a paramount factor. The key idea in this paper is to propose an elliptical rotor scraper pump (ERSP) in order to address the imperfections and defects of traditional volume pumps, such as their labyrinthine design, low volume utilization efficiency and undesirable sealing performance. The ERSP can dramatically achieve the aims of ameliorating pump structure and improving energy efficiency. One of the Roots pump's rotors or the vane pump's shifting blade is replaced by a swing scraper in the ERSP. It is worth noting that a small swing range is a high-priority feature of the scraper; the scraper not only serves the purpose of dividing the pump cavity and self-sealing the unit but also reduces the noise level when working. A vital function of ERSP is the efficient conversion of mechanical energy into fluid pressure energy by changing the volume. Simultaneously, based on theoretical analysis, the mathematical model of the ERSP is established, and its kinematic characteristics are investigated, using ADAMS to verify the kinematic rationality of the ERSP. Ultimately, the research group manufactured a prototype of the pump, based on the previous simulation results and calculations, and experiments verified the feasibility of the pump. The design and research of the ERSP have essential reference significance for the development of fluid energy machinery technology.

**Keywords:** pumps; kinematic characteristics; structural innovation; ADAMS; prototype testing

## 1. Introduction

### 1.1. Research Motivation

As an essential fluid power device, the pump is a component that uses mechanical energy to raise fluid pressure [1,2]. It is extensively used in machinery, hydraulics, automobiles, and other applications. The pump has long performed an irreplaceable role in human manufacturing and life. Nevertheless, the structure has also changed greatly due to the work requirements of different pumps in various fields. More and more types of pumps, designed by various research teams and companies, are currently used in different industries [3]. However, at present, most pump designs have numerous issues, such as a complex structure, not being compact enough, a low displacement rate per unit volume, lack of reliable self-sealing, assembly difficulties, and so on [4]. As such, proposing a new pump that can solve the above problems is critical and has important significance and potential.

### 1.2. Literature Review

The rational utilization and conversion of energy is the focus of energy engineering [5–8], including the world's mainstream pumps, gear pumps, Roots pumps, centrifugal pumps, blade pumps, etc. The working principle of most of these pumps is to change

the fluid volume to achieve fluid pressure [9]. Existing pumps, nevertheless, also have numerous problems and shortcomings that affect the pump's applicability. During the operation of a Roots pump, the meshing of the internal rotor is periodic, which reduces the flow field stability [10]. The surface shape of the Roots pump rotor is complex, which brings considerable difficulties in manufacture and installation. Chen et al. adopted an evolutionary structural optimization method to optimize the rotor design of the Roots pump, and the safety and stability of the optimized structure were significantly improved [11]. When a gear pump is functioning, the essence of its gear rotor engagement is collision contact, which cannot avoid vibration and other problems, resulting in a certain degree of leakage [12]. In addition, low efficiency, loud noise in working, and ease of wear are the gear pump's shortcomings. Wang et al. found effective measures to control gear pump noise and relieve trapped oil when analyzing external gear pump vibration and evaluating pump body design optimization [13]. When a blade pump is functioning, it has high requirements regarding the pressure on both sides of the blade and faces the problems of a short working life and small flow [14]. The rotating speed range of a blade pump is rigorous, and the cleaning degree of the fluid has a high requirement, so the blade pump is at a disadvantage in its scope of application [15]. Yang et al. adopted a new transition curve after considerable testing, which dramatically reduced the noise problem of traditional vane pumps [16].

Numerous researchers and teams have conducted much exploration and analysis of the structural innovation of pumps. Due to the differing working requirements of pumps in different fields, the traditional pump structure is unable to meet all these needs. Zhang et al. used AMESim to model and simulate a new design of variable oil pump. Subsequently, the redundant forces in the mechanical structure were analyzed using ADAMS and improvement directions were proposed [17]. Li et al. proposed a new type of hydraulic pumping unit. This new pump achieves continuous pumping, improves the pumping rate, and offers remarkable energy-saving performance within a symmetrical structure [18]. Hsieh et al. invented a new elliptical curve that can be used as a reference for pump or rotary fluid machinery design [19]. Shim et al. developed a new type of volumetric rotary pump that offers lower vibration, less power loss and a higher flow rate compared with traditional pumps and is suitable for high-viscosity fluids [20]. Choi et al. proposed an improvement direction for centrifugal pumps, based on the performance analysis of an ultra-low specific speed centrifugal pump, and verified its application value [21]. Cheng et al. analyzed the influence of various factors on pump efficiency, in view of the low efficiency of traditional oil pumps, and then developed a small-displacement oil pump. Their experiment proved that their oil pump offers good energy-saving ability and efficiency [22]. In order to improve the volumetric efficiency of the axial piston pump, Zhang et al. proposed a novel stacked-roller 2D piston pump [23]. Guan et al. designed a new spherical pump and carried out kinematic modeling. The spherical pump has many advantages and can be used in many applications [24]. Saputra et al. has developed a pump powered by solar cells that dramatically reduces the cost of domestic pumping needs [25].

Most of the above studies improve upon the original pumps, and their research results have not solved all existing problems. A new kind of elliptical rotor scraper pump (ERSP) is presented in this paper. The ERSP uses a swing scraper instead of a moving blade to realize pressure for self-sealing, forming a novel fluid pump without a valve, increasing work efficiency.

Kinematics research is one of the indispensable steps in investigating and developing a mechanical device. Only when a product meets the requirements of kinematics can it continue to be developed. Sun et al. conducted a kinematic simulation of a pointing mechanism, based on ADAMS, and verified the correctness of kinematic analysis [26]. Yuan et al. proposed a single-stage horizontal self-priming pump system. They conducted a kinematics simulation analysis on its modeling, laying a foundation for the next prototype's production and testing [27]. Battarra et al. carried out a kinematic analysis of a vane CAM ring mechanism for a balanced vane pump. Through parametric analysis, the influence

of different design parameters on the kinematics of the blade CAM ring mechanism was determined [28].

Based on the preceding research and profound consideration of the application and optimization of previous pumps, the ERSP offers tremendous research value. The exploration of this paper is of great significance to the development of the ERSP and has reference value to the improvement of fluid power machinery technology.

### 1.3. Challenges and Problems

Considerably more work will need to be completed to improve the structural innovation and technical capability of conventional positive displacement pumps. Firstly, in recent years, most of the treatises published on pump design innovation and analysis did not fully consider the feasibility of the structure. At the same time, there are not many research teams that are in a position to build prototypes. Secondly, pump research and exploration entail a wide range of aspects, requiring researchers to have engineering design experience. In the current research, the simulation processes are all dependent on related software; as a result, there are specific errors and hysteresis. Finally, the pump is not a device that plays a solo role; the improvement and progress of related technology still need the joint efforts of enterprises and research teams involved in the field.

### 1.4. Contributions of This Work

The major contributions of this paper are as follows:

- This paper proposes a new type of elliptical rotor scraper pump (ERSP) to realize the continuous output of high-pressure fluid.
- The ERSP's structure adopts a swing scraper to divide the pump cavity. This arrangement subverts the design concept of the traditional pump structure.
- Through mathematical modeling, parameter setting and kinematic simulation analysis, the workability of the ERSP is preliminarily verified.
- Based on the current research, the research team developed a prototype to verify the feasibility of the ERSP through practical experiments.

### 1.5. Organization of the Paper

The other chapters of this paper are as follows. Section 2 introduces the structure and working principle of the ERSP. In Section 3, the mathematical model of the pump is established, and related parameters are set up. The kinematic characteristic of the ERSP is analyzed in Section 4, which includes steady-state analysis and followability analysis. In Section 5, a prototype based on R&D is tested. Section 6 summarizes the content of this paper and gives future prospects and suggestions for future work.

## 2. Structure and Working Principle

### 2.1. Pump Structure

The structure of the ERSP is demonstrated in Figure 1, detailing the components that enable the conversion of fluids from low pressure to high pressure. The swing scraper divides the pump cavity into high-pressure and low-pressure cavities, so the ERSP can realize a high-efficiency output by fluid pressure self-sealing. The rotor in the pump follows an elliptical line, which enhances the volume utilization rate and makes it easier to keep the scraper close to the rotor. A compact internal structure and the uncomplicated replacement of parts are the core characteristics of the novel pump. Compared with a traditional pump, the novel pump technology has excellent inheritance characteristics. The ERSP's production process is simple and offers good application value in the future.

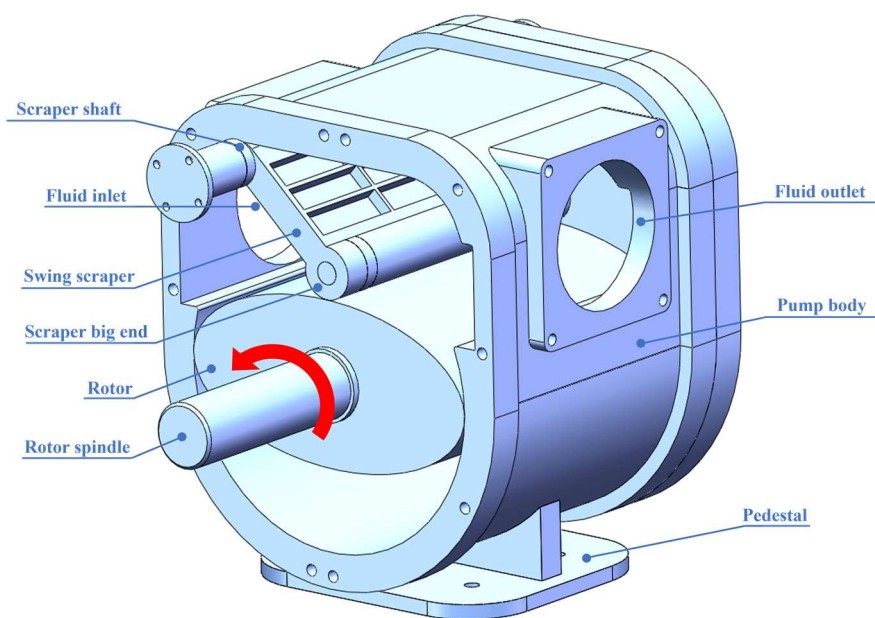

**Figure 1.** The structure of the ERSP.

The main components of the ERSP are an elliptical rotor, rotor spindle, swing scraper, scraper big end, scraper shaft, torsion spring, location pin, deep groove ball bearing, bearing sleeve, bearing shaft, fluid inlet, fluid outlet, pedestal, pump body, flange, and double-lip seals. Part of the detailed structure is illustrated in Figure 2.

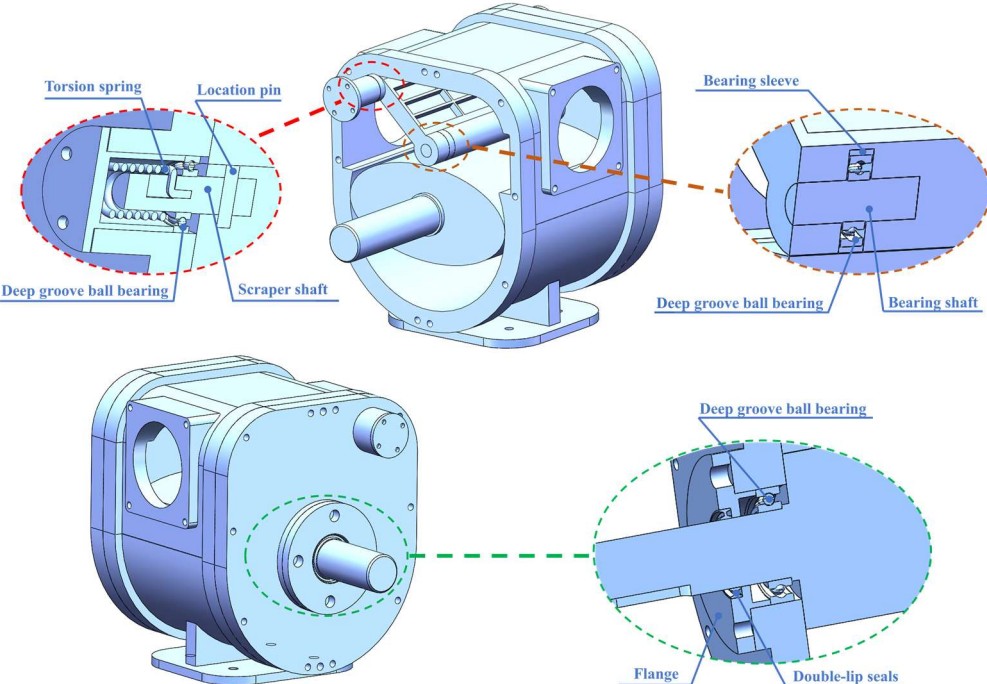

**Figure 2.** Detail of the pump part of the structure.

The pump body and the end cap are fixed by bolts and positioning pins. Large gaps should be avoided between the contact surfaces of each part to reduce internal fluid leakage. The rotor shaft does not move with respect to the elliptical rotor. This design is convenient for casting and also solves the problems of vibration and poor sealing caused by traditional key matching. The swing scraper big end is in close contact with the surface of the rotor, and elliptical lines are applied to the rotor design to achieve synchronous and stable rotation

of the scraper when the elliptical rotor rotates. The swing scraper big end bearing shaft is supported in a deep-groove ball bearing and bearing sleeve; the purpose is to achieve rolling friction between the rotor and scraper, reducing the impact and loss caused by sliding friction. One end of the torsion spring is fixed onto the scraper shaft, the other end is fixed onto the outer flange. The role of the torsion spring is to provide extra torque to keep the swing scraper in contact with the elliptical rotor. There are shoulders on the scraper shaft and the rotor shaft to avoid contact wearing of the bearing outer ring. The clearance between the swing scraper, elliptical rotor and end face can be adjusted by setting a flange. A double-lipped sealing ring is arranged at the flange of the rotor shaft to ensure a good seal in the ERSP unit. The ERSP offers a compact structure, low installation difficulty and good sealing performance. This design of synchronous rotation of rotor and scraper is very uncommon in existing fluid pump research, so it has excellent research value and development prospects.

### 2.2. Working Principles of the ERSP Pump

The ERSP can convert the input of mechanical energy into the output of fluid pressure energy. When the equipment is functioning, the external power source turns the rotor shaft, then the elliptical rotor rotates counterclockwise to drive the scraper to rotate synchronously. As shown in Figure 3, as the rotor rotates, the volume of the low-pressure cavity increases, and low-pressure fluid enters the low-pressure cavity from the fluid inlet. While the rotor is rotating, a local sealing chamber is formed between the rotor and the pump body, and the fluid in the sealing chamber is neither compressed nor expanded. As the high-pressure cavity is squeezed by the rotor, the volume becomes smaller, and the high-pressure fluid is expelled from the fluid outlet. As the elliptical rotor continues to rotate, the low-pressure fluid is constantly sucked into the low-pressure cavity, and the high-pressure fluid is constantly forced out of the high-pressure cavity. The external mechanical energy is continuously transformed into fluid pressure energy.

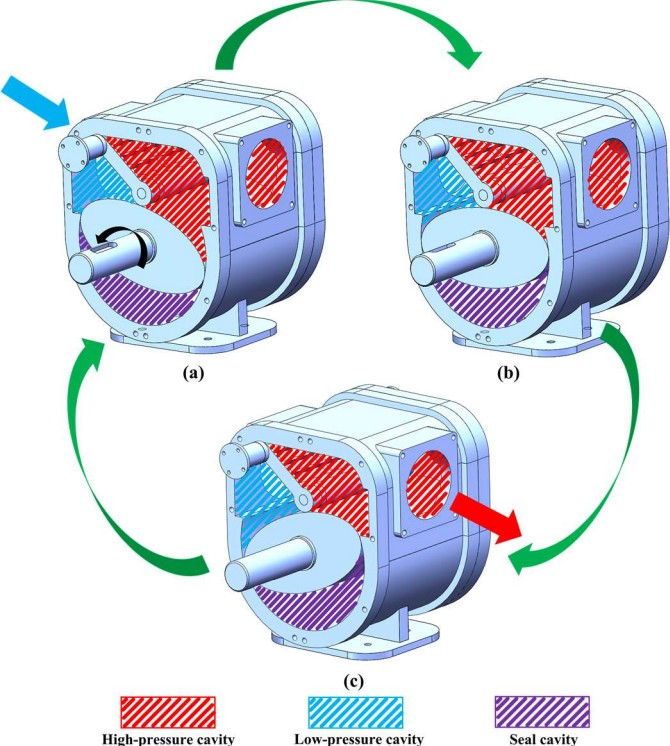

**Figure 3.** Working principle. (**a**) Admission state (**b**) Intermediate state (**c**) Exhaust state.

When the ERSP begins working, the swing scraper can be kept in contact with the rotor under the action of a torsion spring. With the continuous formation and output of high-pressure fluid, the fluid in the high-pressure cavity also produces fluid pressure on the side of the swinging scraper. The double action of the fluid and the torsion spring ensures that the scraper can swing synchronously with the rotor. The pressure difference between high-pressure and low-pressure cavities is large, which further improves the sealing performance between the scraper and the rotor. The output pressure of ERSP equals the pressure difference between the high-pressure and the low-pressure cavities. The pressure difference ranges from 0.02 Mpa to 0.2 Mpa.

Figure 4 is a diagram of the working principle of a Roots pump. In contrast to the ERSP, Roots pumps rotate in different directions with two rotors. The Roots pump also uses volume changes to convert low-pressure fluid into high-pressure fluid. However, the rotors of the Roots pump are periodically engaged, which reduces the stability of the flow field. When the Roots pump is working, the collision between the two rotors is frequent, which can cause serious wear and noise problems. In addition, Roots pump rotor processing is complicated and high in cost.

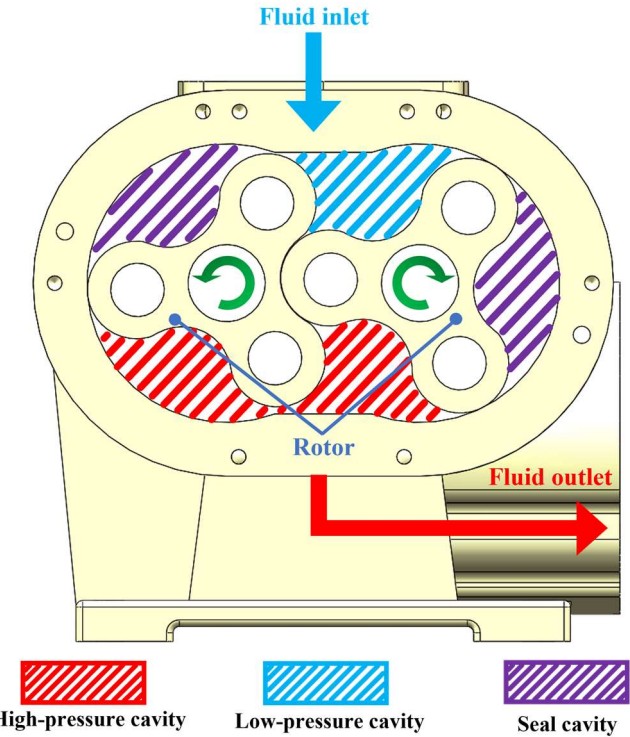

**Figure 4.** Working principle of the Roots pump.

## 3. Mathematical Model and Related Parameters

In order to facilitate the subsequent kinematic research of the ERSP, this chapter establishes the mathematical model of the novel pump. The relevant parameters are determined by theoretical analysis and machining theory. The synchronous motion of the scraper and rotor is further estimated by data calculation.

### 3.1. Mathematical Model

In this paper, the elliptical rotor and swing scraper are key components of the novel device. Moreover, the synchronous movement between the scraper and the rotor is also at the core of the working principle of the pump. Therefore, analysis of the synchronous motion between the swing scraper and the elliptical rotor will be key to the subsequent kinematic analysis [29,30]. This section will establish the mathematical model for synchronous motion between the rotor and the scraper.

As expressed in Figure 5, Figure 5a is a diagram showing the initial state of the ERSP, when the elliptical rotor is horizontal. Figure 5b is a diagram of the limit state of the ERSP when the scraper is at the maximum angle. Setting the center of the rotor axis as the origin of the coordinate axis, $O$, we take the straight line of the long axis of the elliptical rotor section as the $X$-axis and the straight line of the short axis as the $Y$-axis, and then establish the rectangular coordinate system. We then set the center of the scraper shaft as point $P$. Let the center of the scraper big end in Figure 5a be $A_1$ and that in Figure 5b be $A_2$. We then connect $OA_1$, $OA_2$, $PA_1$ and $PA_2$. If Figures 5a,b are placed in the same rectangular coordinate system, the relationship between the maximum rotation angle of the scraper and the rotation angle of the elliptical rotor can be analyzed.

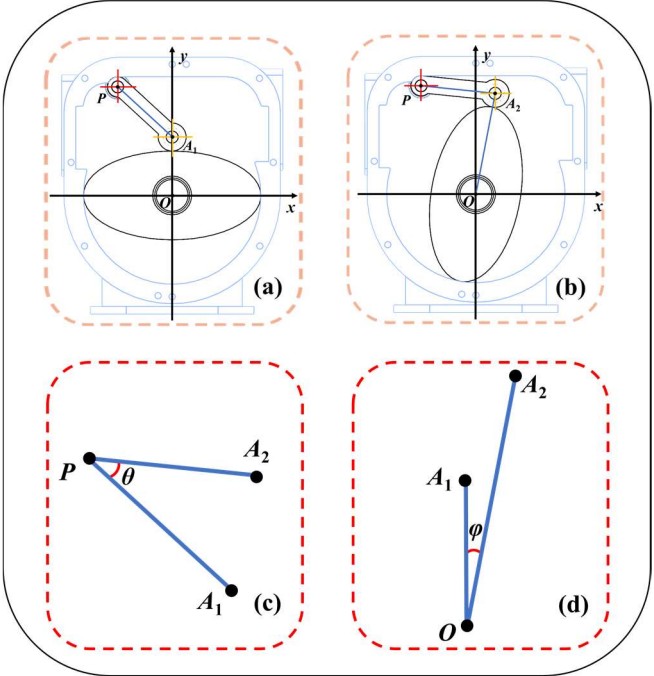

**Figure 5.** Auxiliary diagram for mathematical modeling. (**a**) The initial state of the ERSP, (**b**) rotation limit of the ERSP, (**c**) scraper rotation angle, (**d**) rotor rotation angle.

Figure 5c is the coplanar graph of $PA_1$ and $PA_2$ in the same coordinate system, and Figure 5d is the coplanar graph of $OA_1$ and $OA_2$ in the same coordinate system. Let the angle between $PA_1$ and $PA_2$ be $\theta$, and let the angle between $OA_1$ and $OA_2$ be $\varphi$. Then, the maximum rotation angle of the scraper is $\theta$. When the scraper turns the maximum angle, $\theta$, the elliptical rotor turns to the angle $\pi/2 - \varphi$. Let the coordinates of $O$ be $(0, 0)$, $P$ $(m, n)$, $A_1$ $(x_1, y_1)$, and $A_2$ $(x_2, y_2)$. The $A_1$ and $A_2$ coordinates can be determined by the following two equations:

$$\begin{cases} x_1 = 0 \\ y_1 = OA_1 \end{cases} \tag{1}$$

$$\begin{cases} x_2{}^2 + y_2{}^2 = OA_2{}^2 \\ (x_2 - m)^2 + (y_2 - n)^2 = PA_2{}^2 \end{cases} \tag{2}$$

After the $A_1$ and $A_2$ coordinates are worked out, then:

$$\begin{cases} k_{OA2} = \frac{y_2}{x_2} \\ k_{PA1} = \frac{y_1 - n}{x_1 - m} \\ k_{PA2} = \frac{y_2 - n}{x_2 - m} \end{cases} \tag{3}$$

where $k_{OA2}$, $k_{PA1}$ and $k_{PA2}$ are the slopes of $OA_2$, $PA_1$ and $PA_2$, respectively.

$$\tan \alpha = \frac{|k_1 - k_2|}{1 + k_1 \cdot k_2} \tag{4}$$

The above formula is the tangent formula of the rectangular coordinate system. Here, $k_1$ and $k_2$ are the slopes of two intersecting lines, and $\alpha$ is the included angle of the two lines. Using the tangent formula, we can establish the following relationship:

$$\begin{cases} \theta = \arctan\frac{|k_{PA1} - k_{PA2}|}{1 + k_{PA1} \cdot k_{PA2}} \\ \varphi = \arctan\frac{1}{k_{OA2}} \end{cases} \tag{5}$$

During the operation of the ERSP, the external power drives the rotor shaft to rotate. The rotor speed $n$ (r/min) can be regarded as a constant value, except for the sudden increase of speed caused by external speed regulation. Therefore, there exists the following relationship between the angle $\beta$ (deg) and the speed $n$ (r/min) of the rotor in a certain time frame of $\Delta t$ (s):

$$\beta = 6 \cdot n \cdot \Delta t \tag{6}$$

### 3.2. Basic Parameter Setting

Reasonable parameter setting is significant in mechanical research. In this section, the basic parameters of the ERSP are set up, based on current machining theory and traditional pumps. The swing scraper and elliptical rotor are made from aluminum alloy. Other basic parameters are indicated in Table 1.

**Table 1.** Basic parameters of the ERSP.

| Parameters | Value |
| --- | --- |
| Long axis of elliptical line | 228 mm |
| Short axis of elliptical line | 114 mm |
| Length of the $PA_1$ | 95.1804 mm |
| Thickness of swing scraper | 228 mm |
| Thickness of elliptical rotor | 228 mm |
| Diameter of scraper big end section circle | 36 mm |
| Elliptical rotor speed ($n$) | 300 r/min |
| Torsion spring stiffness | 0.0565 N·m/deg |

According to the design of the ERSP and the setting of basic parameters, the mathematical model in Section 3.1 can be substituted in real numbers. The angle of $\theta$ is about 36.723°, and the angle of $\varphi$ is about 10.750°. When the swing scraper turns at the maximum angle of 36.723°, the rotor turns to 79.250°.

### 4. Kinematic Characteristics Analysis

Theoretical analysis and mathematical modeling were carried out, as outlined in the above chapters, and this chapter uses ADAMS to analyze the kinematic characteristics of the ERSP. ADAMS, or the automatic dynamic analysis of mechanical systems, can simulate the statics, kinematics and dynamics of mechanical systems. ADAMS simulations can be used to predict mechanical system performance, motion range, collision detection, peak load and calculate the input load of finite elements. This paper mainly uses the kinematics analysis function of ADAMS.

In this chapter, kinematic characteristics are analyzed from two aspects. Firstly, when the rotor speed and pressure difference are fixed and the scraper big end is always close to the rotor, the kinematic characteristics are analyzed. This aspect is called steady-state analysis. Secondly, the parameters are changed to observe the kinematic characteristics of the swing scraper follow-through. This aspect is called followability analysis.

### 4.1. Steady-State Analysis

In the steady-state analysis, we selected the rotating speed, *n*, of the elliptical rotor as 300 r/min, and the pressure difference between high-pressure and low-pressure cavities is taken as 0.5 atmospheric pressure (about 0.05 MPa). This section takes the horizontal state of the elliptical rotor as the initial state for the simulation and selects the first three periods of the pump to research.

Figure 6 shows the rotation angle curve of the swing scraper, as obtained by the simulation. It is obvious from Figure 6 that when the ERSP completes a cycle, the swing scraper swings up and down twice. Through the analysis of the simulation data, it is known that the maximum angle of swing scraper rotation is 36.765°. The angle obtained by the simulation is approximately equal to the maximum angle of 36.723°, as derived by the mathematical model in Section 3. When the rotation angle of the swing scraper reaches the maximum angle for the first time, the time elapsed is 0.044 s.

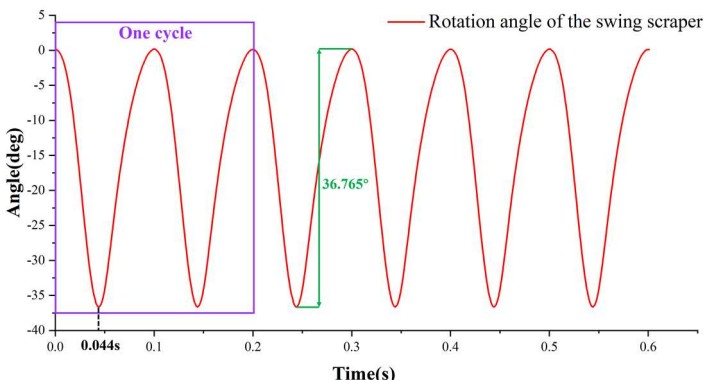

**Figure 6.** The curve of the rotation angle of the swing scraper.

Figure 7 shows the curve of the rotation angle of the elliptic rotor, **as** obtained by the simulation. It is obvious from the figure that when the ERSP completes a cycle, the elliptical rotor rotates once. According to the rotation angle of the swing scraper and elliptical rotor, when the scraper turns the maximum angle of 36.765° for the first time, the scraper turns 79.200°. The rotor rotation angle obtained by the simulation is approximately equal to 79.250°, as derived from the mathematical model in Section 3. A comparison between the simulation and theoretical model is given in Table 2.

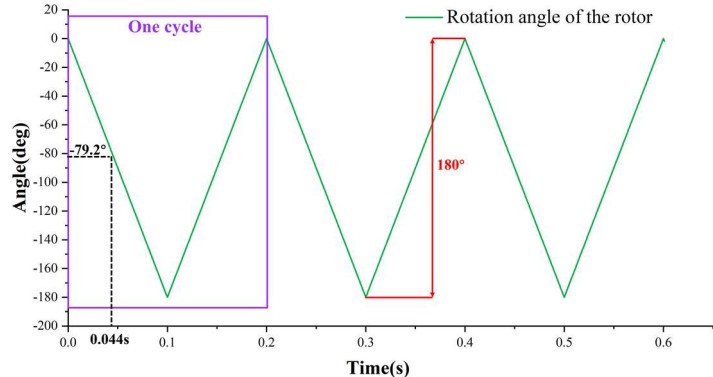

**Figure 7.** The curve of the rotation angle of the rotor.

**Table 2.** Comparison between the simulation and theoretical model.

| Parameters | Theoretical Calculation Value | Simulation Value |
| --- | --- | --- |
| Maximum angle of scraper ($\theta$) | 36.723° | 36.765° |
| Angle of elliptical rotor ($\pi/2 - \varphi$) | 79.250° | 79.250° |

Figure 8 depicts the comparison between the rotation angle of the elliptical rotor and the swing scraper. The angled contrast more clearly reflects the ERSP's working principle. The maximum rotation angle of the swing scraper is slight, which can effectively reduce both noise and impact.

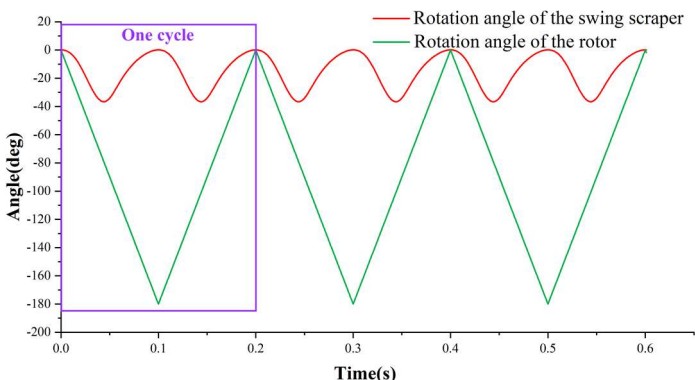

**Figure 8.** Comparison of the rotation angles.

Figure 9 shows the angular velocity curve of the swing scraper. It can be seen from this curve that in the initial three periods, the angular velocity of the swinging scraper also has periodicity. According to the simulation data, the angular velocity of the swing scraper oscillates in the range of 0–24.09 rad/s. After further calculation, the maximum rotation speed of the swinging scraper is about 235.64 r/min. The maximum rotation speed of the swing scraper is less than the rotation speed of the elliptical rotor, which also verifies that under such working conditions, the swing scraper can be stably attached to the elliptical rotor. The above results ensure the ERSP's sealing and working stability.

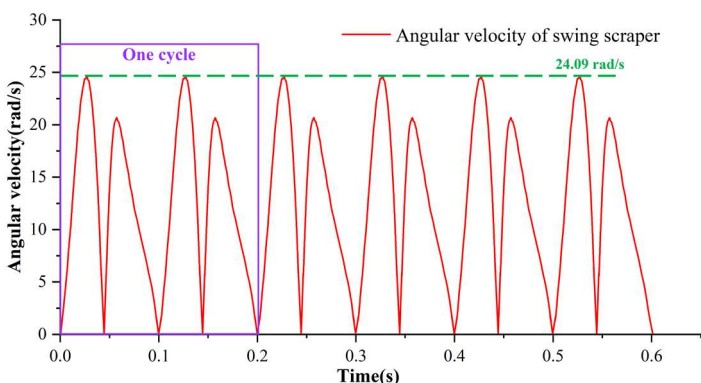

**Figure 9.** The curve of angular velocity of the swing scraper.

The simulated curve of the angular acceleration of the swing scraper is shown in Figure 10. When the pump starts, the rotor rotates to drive the scraper, so the scraper undergoes great acceleration at the start moment. On the whole, the angular acceleration of the swing scraper is also periodic. According to data analysis, the maximum angular acceleration is about 2562.4 rad/s$^2$, and the average angular acceleration is 1410.4 rad/s$^2$.

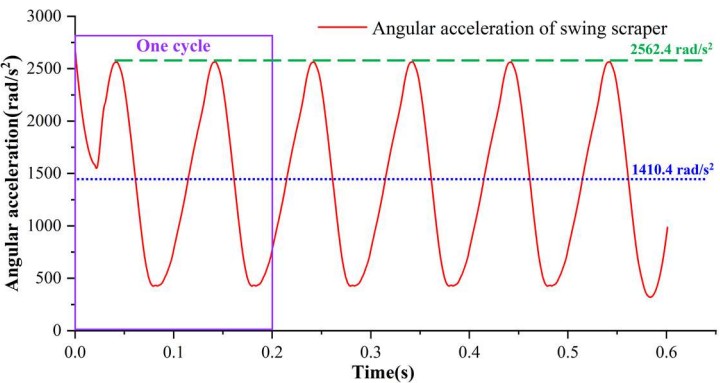

**Figure 10.** The curve of angular acceleration of the swing scraper.

Figure 11 describes the curve of torsion spring deformation established by the simulation. In terms of parameter settings, the pre-torsion of the torsion spring has been set at 15° in this paper. Through data analysis, the maximum deformation of the torsion spring is about 51.760°. Figure 12 shows the torque curve of the torsion spring. Through data analysis, the pre-tightening torque of the torsion spring is set at 830.4 N·mm, and the maximum torque is 2939.7 N·mm. In summary, the torsion spring plays the role of providing torque for auxiliary sealing in the working process of the ERSP. Due to the low torque provided, the efficiency of the ERSP is not reduced during operation.

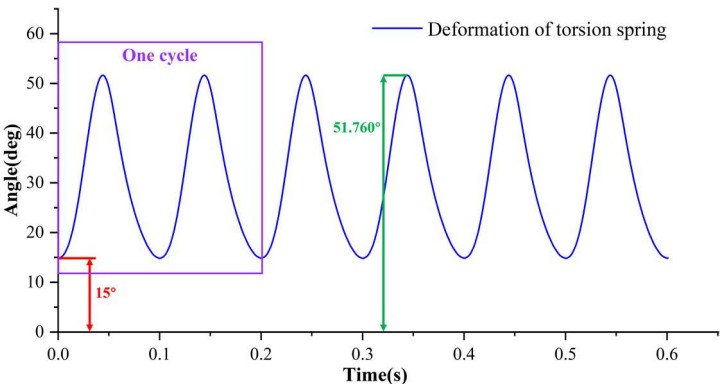

**Figure 11.** The curve of deformation of the torsion spring.

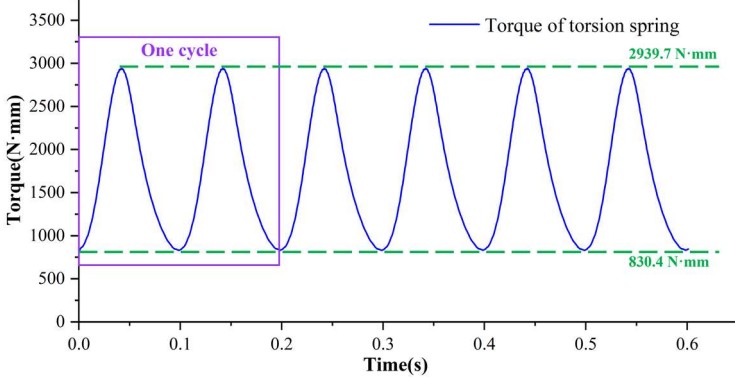

**Figure 12.** The curve of the torque of the torsion spring.

Figure 13 shows the contact force curve between the elliptical rotor and the swing scraper, as obtained by the simulation. The instantaneous direction of the contact force should be perpendicular to the elliptical tangent of the contact point. Through data analysis, the maximum contact force is set at about 1201.2 N.

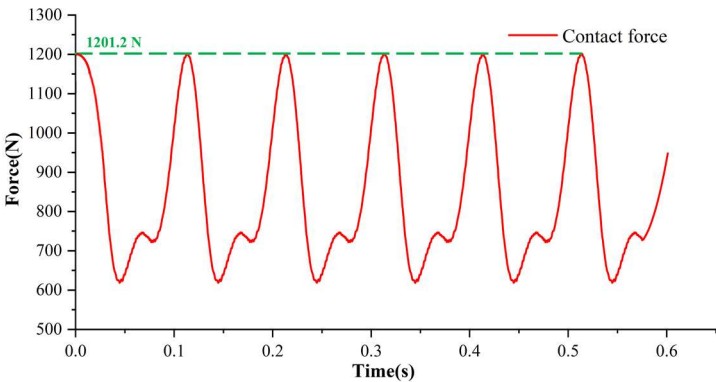

**Figure 13.** The curve of contact force.

*4.2. Followability Analysis*

In the ordinary operation of the ERSP, it is necessary to ensure that the swing scraper big end is always close to the elliptical rotor. Therefore, it is essential to analyze and study the following ability of the swing scraper relative to the elliptical rotor. In engineering practice, in order to make the scraper adapt faster to the rotor speed, we can increase the reverse torque of the swing scraper by adjusting the stiffness of the torsion spring. In addition, the pressure difference between high-pressure and low-pressure cavities is also variable under different working conditions. Therefore, when the above parameters are changed, the maximum speed of the rotor in the normal operation of the ERSP will also change. Figure 14 displays the followability analysis of the surface figure. Figure 14 clearly shows the maximum rotor speed that can make the swing scraper follow the rotor well under different torsion spring stiffness settings and various pressure differences between the high-pressure and low-pressure cavities. In the study, the torsion spring stiffness range is 56.5 N·mm/deg–615.85 N·mm/deg, and the pressure difference is 0.02 MPa–0.2 MPa. The simulation results show that the greater the stiffness of the torsion spring or the pressure difference, the better the following performance of the ERSP. The relevant matching data, obtained from the followability analysis in this section, is of great significance to follow-up research and the prototype testing of the novel pump.

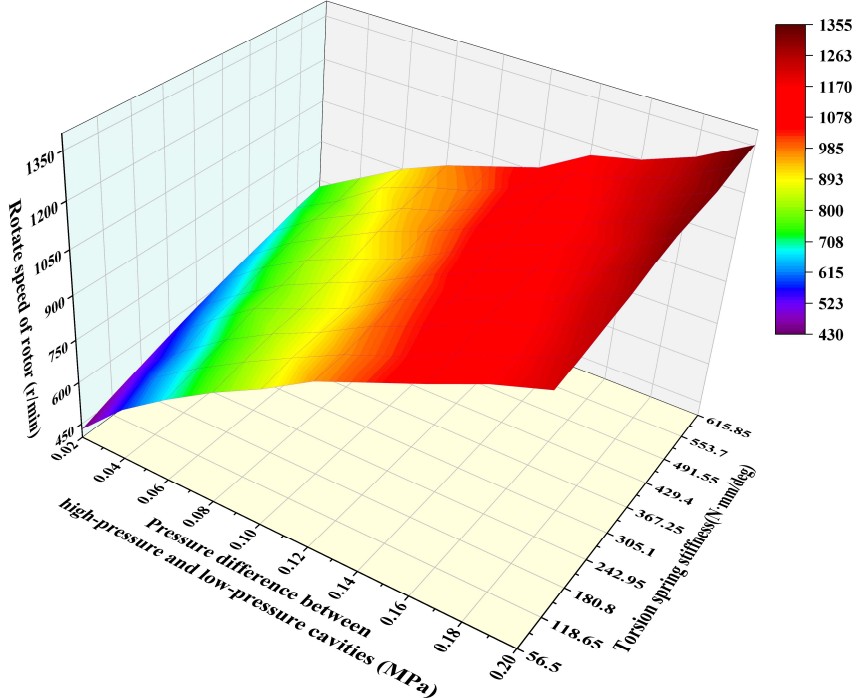

**Figure 14.** Result of the followability analysis.

## 5. Development of Principle Prototype

This research group has carried out an in-depth analysis of the ERSP in terms of theoretical analysis, model building and characteristics research. Building on existing research, a principle prototype pump has been successfully developed. The relevant R&D process is shown in Figure 15. The rotor shaft of the prototype is connected to an AC motor, and the motor is set to output at a constant speed to establish the synchronous movement of the elliptical rotor and the swing scraper. The prototype test proves that when the motor speed is set at 300 r/min, 450 r/min and 600 r/min, respectively, the swing scraper has acceptable following performance and can meet the working requirements.

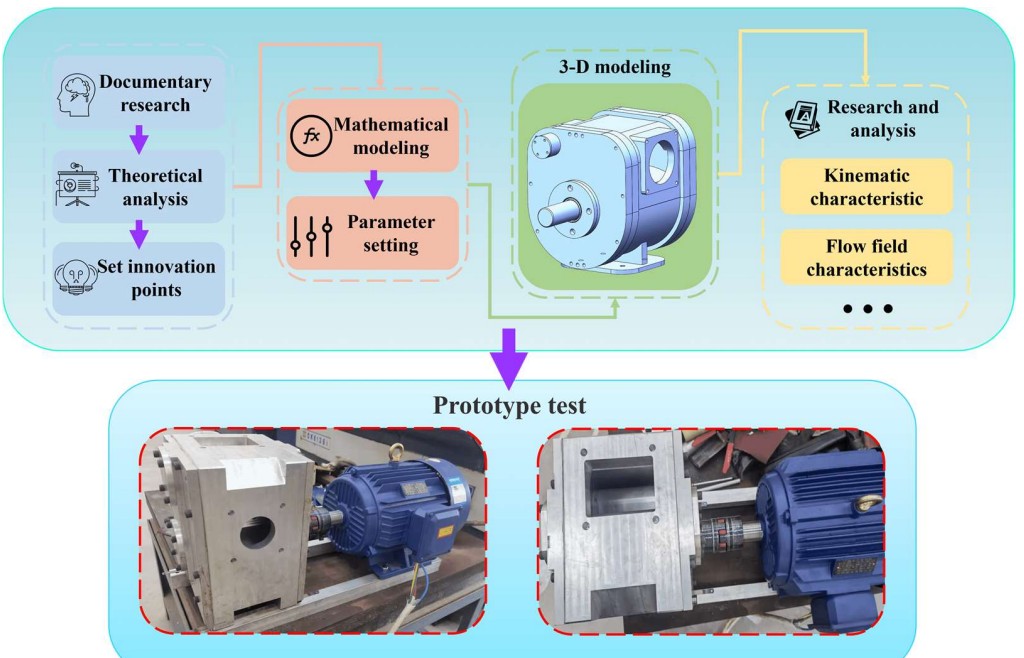

**Figure 15.** R&D process.

## 6. Conclusions and Prospects

It is indisputable that the elliptical rotor scraper pump (ERSP) is presented herein for the first time and its kinematic characteristics have been studied in this paper. The novel pump adopts a swinging scraper instead of the blade and rotor used in a traditional positive displacement pump. By changing the pump cavity volume, the ERSP can convert mechanical energy into fluid pressure energy. The ERSP surmounts the problems of poor sealing performance and the low volume utilization effectivity of traditional positive displacement pumps, and can more comfortably meet the requirements of various fluid pressure outputs. The mathematical model was established through theoretical analysis in the above chapters, and the ERSP's kinematics characteristics were anatomized via ADAMS.

Steady-state analysis and followability analysis are the fundamental components of the analysis of kinematic characteristics. In the steady-state analysis, rotor speed and the pressure difference between high-pressure and low-pressure cavities were taken as fixed values. Through kinematic simulation, the rotation angle and contact force curves were output, which fully verify the kinematic feasibility of the ERSP. In the followability analysis, the maximum rotor speed was set using different rotor speeds and the pressure difference between high-pressure and low-pressure cavities was obtained via simulation. The followability analysis is of great significance to subsequent engineering practice.

It is worth mentioning that the ERSP breaks the traditional concept of positive displacement pump design. The ERSP offers certain advantages in terms of installation layout and sealing and has broad development prospects in the future. In the long run, this

study has momentous reference significance for the improvement and development of fluid machinery and engineering.

**Author Contributions:** Conceptualization, T.Z. and H.Z.; methodology, Z.Z. and H.Z.; software, Z.Z. and J.Y.; validation, Z.Z., Y.C. and J.Y.; formal analysis, Y.J. and D.T.; investigation, Z.Z.; resources, T.Z. and H.Z.; data curation, Z.Z., Y.C. and J.Y.; writing—original draft preparation, Z.Z.; writing—review and editing, Z.Z. and J.Y.; visualization, Z.Z.; supervision, Z.Z. and J.Y.; project administration, T.Z. and H.Z.; funding acquisition, H.Z. All authors have read and agreed to the published version of the manuscript.

**Funding:** This research was funded by the National Natural Science Foundation of China, grant number 52075278, and the Municipal Livelihood Science and Technology Project of Qingdao, grant number 19-6-1-92-nsh.

**Data Availability Statement:** Not applicable.

**Conflicts of Interest:** The authors declare no conflict of interest.

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
