# Peer review of "Modeling, Kinematic Characteristics Analysis and Experimental Testing of an Elliptical Rotor Scraper Pump"

_machines, doi:10.3390/machines10020078_

Round 1

Reviewer 1 Report

This manuscript investigates an elliptical rotor scraper pump (ERSP), by improving the configuration and energy efficiency. The design and study of this work is interesting and novel enough.  However, there are still several concerns in reading through this manuscript. The reviewer suggests a major revision, mandatory.

  1. Whether the proposed elliptical rotor scraper pump is applied in any environment? For example, the marine equipment?
  2. The pressure range of suction and discharge chamber should be presented in Introduction, so that the readers can understand its basic performance.
  3. In paragraphs 2-3 of Page 2, the literature review for structural innovation of pumps is not enough, for example, the spherical pump illustrated below:

Kinematic modeling, analysis and test on a quiet spherical pump[J]. Journal of Sound and Vibration, 2016, 383: 146-155.

  1. In Fig. 1, anticlockwise arrow should be noted on the of rotor firstly, so that the working principle of such pump can be demonstrated clearly based on its overall configuration.
  2. Clearance between the joint of scraper shaft and pump cylinder is the most sophisticated portion, because it may lock (small clearance) or leakage (high clearance) between the suction/discharge chambers, how to tackle these problems? Furthermore, it is a dynamic sealing surface which can be found in Fig. 4 as the scraper rotates continuously.
  3. The sub caption of each figure should be illustrated, for example, Fig .4(a)***, (b)****, (c)****.
  4. There are typos in Eqs. (4) and (5), besides, the formula number is incorrect as well, i.e., three equation (4), it is weird.
  5. In Figs. 11 and 12, the back torque and contact force is up to 3 Nm and 1200 N respectively, therefore durability may be a big issue for this kind of pump.
  6. How the authors to obtain the Fig. 13, whether it is based on theoretical analyses or virtual prototyping, whether it gravity of scraper is considered?
  7. The English in this manuscript should be improved comprehensively.

Reviewer 2 Report

In this study, the authors proposed a new type of elliptical rotor scraper pump (ERSP). Through mathematical modeling, parameter setting and kinematic simulation analysis, the work rationality of the ERSP is preliminarily verified. The topic of the paper is interesting and the manuscript is comprehensive. However, the results need to be more consistent and clearly presents. In addition, the English style and format should be checked. I would like to provide some comments, which may help to improve the manuscripts as below:

  1. The English typing, journal format should be carefully checked. It has some errors. For instance, equations, tables, etc.
  2. The simulation software/ tool  (Adams) needs to be presented in more detail.
  3. About simulation time, how do the authors define the time of one cycle (0.2s)? Does it follow standards? Please clarify.
  4. The results show in a suitable way with well-prepared figures. However, it may be missing the comparison with others types of pumps. In the introduction, the authors told about the limitations of other pumps. Is it possible to provide a comparison with others (same volume flow rate/ same speed)? Please clarify.
  5. The comparison between simulation and theoretical model should be presented in figures/ tables.
  6. In the title, the authors state that "experiment". However, in your results, there were no experimental results. The authors only presented the diagram for modeling a type of pump. This may make readers misunderstand. Please clarify. If the authors did the experiment, please provide the comparison with simulation/ theoretical analysis.

Round 2

Reviewer 1 Report

accept as It is

Reviewer 2 Report

The quality of the manuscript has been improved based on comments. I would like to recommend it for publishing in Machines.